# Are models trained on temporally-continuous data streams more adversarially robust?

**Nathan C. L. Kong**[*]
Stanford University
nclkong@stanford.edu

**Anthony M. Norcia**
Stanford University
amnorcia@stanford.edu

## Abstract

Task-optimized convolutional neural networks are the most quantitatively accurate models of the primate visual system. Unlike humans, however, these models can easily be fooled by modifying their inputs with human-imperceptible image perturbations, resulting in poor *adversarial robustness*. Prior work showed that modifying a model's training objective or its architecture can improve its adversarial robustness. Another ingredient in building computational models of sensory cortex is the training dataset and, to our knowledge, its effect on a model's adversarial robustness has not been investigated. Motivated by observations that newborn chicks (*Gallus gallus*) develop more invariant visual representations when reared with more temporally-continuous visual experience, we here evaluate a model's adversarial robustness when it is trained on a more naturalistic dataset—a longitudinal video dataset collected from the perspective of infants (SAYCam; Sullivan et al., 2020). By evaluating the adversarial robustness of models on 26-way classification of a set of annotated video frames from this dataset, we find that, across multiple objective functions, models that have been pre-trained on SAYCam video frames are more adversarially robust than those that have been pre-trained on ImageNet. Our results suggest that to build models that are more adversarially robust, additional efforts should be made in curating datasets that are more similar to the natural image sequences and the visual experience that infants receive.

## 1 Introduction

Task-optimized deep convolutional neural networks (CNNs) are the most quantitatively accurate models of the human and non-human primate ventral visual stream (Yamins et al., 2014; Khaligh-Razavi & Kriegeskorte, 2014; Schrimpf et al., 2018; Bashivan et al., 2019). However, their image-classification behavior on images that have been modified by non-random, human-imperceptible image perturbations diverges significantly from that of humans. Specifically, these *adversarial perturbations* can cause CNNs to confidently misclassify an image even though they can classify the unperturbed image correctly (Szegedy et al., 2014). We view the problem of building computational models (of the visual system) that are robust to these perturbations through the lens of the goal-driven modeling approach (Yamins & DiCarlo, 2016), which consists of four major components: objective function, model architecture, data stream, and learning rule. Each component can be tweaked in order to better achieve an optimization goal, which, in this work, is adversarial robustness.

Some work focused on reformulating a generic objective function such as the cross-entropy loss in image classification to improve the adversarial robustness of CNNs (e.g., Madry et al., 2018; Ross & Doshi-Velez, 2018; Zhang et al., 2019). Other work focused on architectural modifications to improve adversarial robustness (e.g., Vuyyuru et al., 2020; Dapello et al., 2020). To our knowledge,

---

[*]To whom correspondence should be addressed.

3rd Workshop on Shared Visual Representations in Human and Machine Intelligence (SVRHM 2021) of the Neural Information Processing Systems (NeurIPS) conference, Virtual.

however, the impact of a model's *training data* on its adversarial robustness has not been investigated before and therefore is the focus of the work presented here.

We were motivated to study the effects of the data stream on robustness because of observations in controlled-rearing studies of newborn chicks (*Gallus gallus*; Wood & Wood, 2016; Wood et al., 2016; Wood & Wood, 2018). Specifically, the authors found that newborn chicks reared with more temporally-continuous visual experience were better able to generalize to novel viewpoints of objects and to recognize familiar objects than chicks reared with less temporally continuous visual experience. As the visual experience acquired via training on ImageNet (Deng et al., 2009) is highly unrealistic (developmentally) relative to that received by human infants due to, for example, the sheer number of category labels or the lack of temporal continuity in the data, we reasoned that training models on a more developmentally-realistic data distribution (SAYCam; Sullivan et al., 2020) could lead to visual representations that are more robust to input perturbations.

Our work is closely related to that of Orhan et al. (2020) and of Zhuang et al. (2021), who both used the same infant head-cam video dataset (SAYCam) for training models. Orhan et al. (2020) showed that models can learn visual representations useful for basic downstream tasks by training them on head-cam video frames using self-supervised techniques. Similarly, Zhuang et al. (2021) showed that models trained using the head-cam dataset are also competitive models of the ventral visual stream. In both cases, however, the model's adversarial robustness was not evaluated.

## 2 Methods

**Dataset**    We used video frames extracted from a subset of the SAYCam infant head-cam dataset (Sullivan et al., 2020), which is a longitudinal, egocentric dataset of videos captured using head cameras placed on three infants. Specifically, we used the videos from child S and from child Y and divided them into approximately five-minute episodes. Video frames were then extracted from these episodes at five frames per second, as in Orhan et al. (2020). In order to evaluate models trained using these images or using ImageNet on a downstream classification task and on adversarial robustness, we used the annotated video frames from child S provided by Orhan et al. (2020). In the annotated dataset, there are approximately $58\,000$ annotated video frames, where $50\%$ of those belonged to the train set and the remaining $50\%$ belonged to the validation set. Each of these images belonged to one of 26 categories, including cat, car, and basket. Sample images from this annotated subset of video frames are shown in Figure 3 (found in Appendix A.1).

**Model architecture**    We fixed the model architecture to be ResNet-18 (He et al., 2016), as it has been shown to be one of the best models of the ventral visual stream when trained on supervised image classification (Schrimpf et al., 2018). This architecture has also been shown to be a highly accurate model of the ventral visual stream when trained in a self-supervised manner (Zhuang et al., 2021).

**Objective functions**    For models trained using ImageNet, three objective functions were used: cross-entropy loss for supervised learning (from PyTorch's model zoo; "Supervised ImageNet"), simple siamese representation learning for self-supervised learning (SimSiam; Chen & He, 2021), and momentum contrast for self-supervised learning (MoCov2; Chen et al., 2020) ("Contrastive ImageNet"). SimSiam and MoCov2 are two state-of-the-art contrastive learning algorithms that do not need memory banks or large batches during training. For the supervised ImageNet model, we also trained a variation of it that used the same image augmentations as those used in the contrastive learning algorithms ("Supervised ImageNet++"). As a strong control model, we also used a ResNet-18 that was adversarially trained on ImageNet to be robust to adversarial perturbations, $\boldsymbol{\delta}$, of size at most $\|\boldsymbol{\delta}\|_\infty \leq 1/255$ ("Adversarial ImageNet"; Salman et al., 2020).

For models trained using video frames from the SAYCam dataset, three self-supervised objective functions were used: temporal classification ("Temporal Classification SAYCam"), SimSiam, and MoCov2 ("Contrastive SAYCam"). The temporal classification objective encourages models to be able to classify video frames according to the five-minute episode to which they belong (Orhan et al., 2020). For the contrastive objective functions, we also trained models using a slight modification, where two temporally-adjacent video frames were considered positive examples, on top of image augmentations being positive examples for an image (temporal contrastive learning, "Temporal Contrastive SAYCam"; Orhan et al., 2020).

**Model training** For the temporal classification and the temporal contrastive learning objectives, we used similar hyperparameters and the same image augmentation scheme as in the code base of Orhan et al. (2020). Image augmentations used when training models on the SAYCam dataset using SimSiam and MoCov2 are the same as those used by Chen & He (2021) and by Chen et al. (2020). Additional model training and image augmentation details can be found in Appendix A.4.

**Linear classifier training** Models trained using each of the dataset-objective-function combinations were evaluated on their 26-way linear classification performance on the set of annotated video frames from child S, similar to the procedure of Orhan et al. (2020). If the model had been trained in a supervised manner (e.g., supervised ImageNet), its final fully-connected (`fc`) layer was removed and replaced with a linear readout of dimensions $512 \times 26$ (i.e., the linear readout was placed post-average pool in ResNet-18). For self-supervised models, an `fc` layer of the same dimensions as above was appended to the end of those models, which do not have `fc` layers. Additional model training and image augmentation details can be found in Appendix A.4.

**Adversarial robustness evaluation** Adversarial robustness was evaluated on the 26-way linear classification task so that each of our models pre-trained (in a supervised or a self-supervised manner) on either ImageNet or SAYCam had a linear readout layer for the 26-way class predictions. We focused primarily on "white-box" adversarial attacks and used projected gradient descent (PGD; Madry et al., 2018) to find the image perturbations that maximized the cross-entropy loss (i.e., untargeted attacks). Adversarial examples were generated using 20 steps of PGD with step sizes of $\varepsilon \times 2/20$, where $\varepsilon$ is the maximum allowed "size" of the perturbation. The sizes of the perturbations were measured using the $\ell_\infty$-, $\ell_2$-, and $\ell_1$-norm. Additional details on the size constraints can be found in Appendix A.5.

## 3 Results

### 3.1 Adversarial Examples

Using a ResNet-18 pre-trained in a supervised manner on ImageNet, along with its linear readout trained to perform the 26-way classification task, we generated a few example adversarial images for demonstration, shown in Figure 4 (found in Appendix A.2). The adversarial images were generated using perturbations constrained by the $\ell_\infty$-norm and $\varepsilon = 1/1020$ (i.e., $\|\boldsymbol{\delta}\|_\infty \leq 1/1020$, where $\boldsymbol{\delta}$ is the perturbation). This provides additional evidence that human-imperceptible image perturbations can drastically affect a model's behavior.

### 3.2 Adversarial Accuracy

Using models trained on either ImageNet or on the SAYCam dataset, we trained 26-way linear classifiers on top of the (frozen) pre-trained backbones to perform classification of annotated video frames from child S. Here we directly compare the clean and the adversarial accuracy on the 26-way classification task.

Table 1 shows the accuracy on the classification task when the images have or have not been affected by relatively weakly-constrained adversarial perturbations. We found that ImageNet-trained models have comparable accuracy on unperturbed images with SAYCam-trained models (Table 1; **Clean Accuracy**), but have worse accuracy on images that have been weakly adversarially perturbed. When comparing "SimSiam ImageNet" with "SimSiam Y", where *only* the training dataset is different (same architecture and objective), we find that even with comparable clean accuracy, "SimSiam Y" exhibits higher adversarial accuracy for all perturbation types. Furthermore, comparing "Contrastive SAYCam" models with "Temporal Contrastive SAYCam" models, where the training set (and the architecture) was held fixed and only the objective changed so that temporally-adjacent video frames were treated as positive examples, we found that models trained using temporal contrastive learning had higher adversarial accuracy than that of models trained using vanilla contrastive learning (for example, compare "MoCov2 Y" with "MoCov2-Temporal Y" in Table 1).

Across all the tested perturbation sizes, we observed that models trained on either child S's or child Y's video frames exhibited higher adversarial robustness than those *not* explicitly trained to be adversarially robust on ImageNet, shown in Figure 1 (compare all colors except for black). Explicit

adversarial training on ImageNet led to the most adversarially robust models. Furthermore and as expected, for each norm constraint, model performance on the linear classification task plummets to zero as the maximum allowable perturbation size increases, suggesting consistent adversarial vulnerability across models. Overall, "Supervised ImageNet" and "Supervised ImageNet++" performed the worst, with their accuracy dropping off the fastest (bottom two curves in Figure 1).

Table 1: **Clean and adversarial accuracy for each model (ResNet-18) trained on each dataset-objective-function combination on the** $26$**-way classification of the annotated video frames from child S.** Each column is associated with a different type of constraint applied on the perturbation. S and Y in the model names refer to which dataset from SAYCam the model was trained on. For example, "SimSiam Y" means that the model was trained on child Y videos using the SimSiam objective and "SimSiam-Temporal Y" means that the model was trained on child Y videos using the "temporal" version of the SimSiam objective (Orhan et al., 2020). "Adversarial ImageNet" denotes a ResNet-18 architecture adversarially trained on ImageNet to be robust to perturbations, $\boldsymbol{\delta}$, of size at most $\|\boldsymbol{\delta}\|_\infty \leq 1/255$ (Salman et al., 2020). The performance values reported here are the accuracy for a given size constraint for each norm. For the "Random" model, a 26-way linear readout was trained on top of a randomly initialized ResNet-18.

| Model | Clean Accuracy | $\|\boldsymbol{\delta}\|_\infty \leq 1/1020$ | $\|\boldsymbol{\delta}\|_2 \leq 0.15$ | $\|\boldsymbol{\delta}\|_1 \leq 40$ |
|---|---|---|---|---|
| Supervised ImageNet | 55.8% | 2.7% | 8.7% | 8.1% |
| Supervised ImageNet++ | 48.4% | 1.4% | 3.5% | 3.0% |
| SimSiam ImageNet | 46.6% | 12.5% | 18.5% | 17.5% |
| MoCov2 ImageNet | 54.4% | 6.7% | 12.4% | 11.4% |
| Adversarial ImageNet | 56.4% | 51.6% | 52.8% | 51.5% |
| SimSiam S | 37.9% | 16.4% | 21.0% | 19.8% |
| SimSiam-Temporal S | 35.4% | 21.8% | 25.9% | 24.8% |
| SimSiam Y | 46.7% | 20.9% | 27.7% | 26.3% |
| SimSiam-Temporal Y | 46.6% | 24.6% | 31.0% | 29.7% |
| MoCov2 Y | 51.6% | 25.0% | 31.5% | 29.3% |
| MoCov2-Temporal Y | 54.0% | 31.3% | 37.9% | 36.5% |
| Temporal Classification S | 63.1% | 22.5% | 32.2% | 30.1% |
| Temporal Classification Y | 54.4% | 19.4% | 27.6% | 25.7% |
| Random | 28.3% | 11.3% | 17.7% | 18.4% |

### 3.3 Spatial Frequency Tuning Analysis

We next asked whether there were differences in spatial frequency tuning preferences between models that were trained on ImageNet and those that were trained on SAYCam. We investigated this property because prior work has suggested that the brittleness of these models can be associated with their sensitivity to image features of high spatial frequencies (Yin et al., 2019; Wang et al., 2020; Kong et al., 2021). To generate spatial frequency tuning curves for each artificial neuron, we presented each model with a set of fixed-size Gabor patches that varied in spatial frequency, phase, and orientation and "recorded" activations for each neuron in the `maxpool` layer of ResNet-18. Each tuning curve (for each neuron) was then summarized by the spatial frequency at which the tuning curve reached its maximum value, resulting in a distribution of preferred spatial frequencies for the neurons in the `maxpool` layer.

The preferred spatial frequency distributions for the models trained on ImageNet and on SAYCam are shown in Figure 2. Consistent with other findings, we found that a large proportion of artificial neurons in less adversarially robust models were tuned to high spatial frequencies (Figure 2; orange). In contrast, models that were more adversarially robust had a relatively smaller proportion of artificial neurons tuned to the highest spatial frequencies (Figure 2; red and purple) and more neurons were tuned to the middle spatial frequencies. Furthermore, training a model on ImageNet in a supervised manner with more image augmentations ("Supervised ImageNet++"), including Gaussian blur, did not reduce the proportion of artificial neurons tuned to high spatial frequencies (Figure 5, found in Appendix A.3).

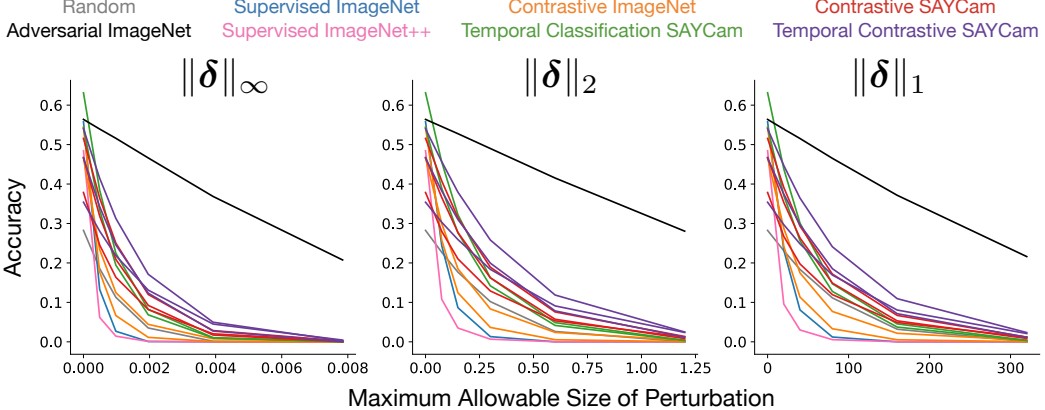

Figure 1: **Adversarial accuracy on the 26-way classification of annotated frames from child S as a function of perturbation size for each norm.** For each norm (shown in each subplot), as the perturbation size increases, the accuracy of all the models is driven to zero. We refer the reader to Appendix A.5 for further details on the size constraints used. We find that models pre-trained on the head-cam dataset (using videos from child S or from child Y; green, red, purple) have higher adversarial accuracy than that of models pre-trained on ImageNet (blue, pink, orange). All the models, however, are less robust than a model that was adversarially trained on ImageNet (black).

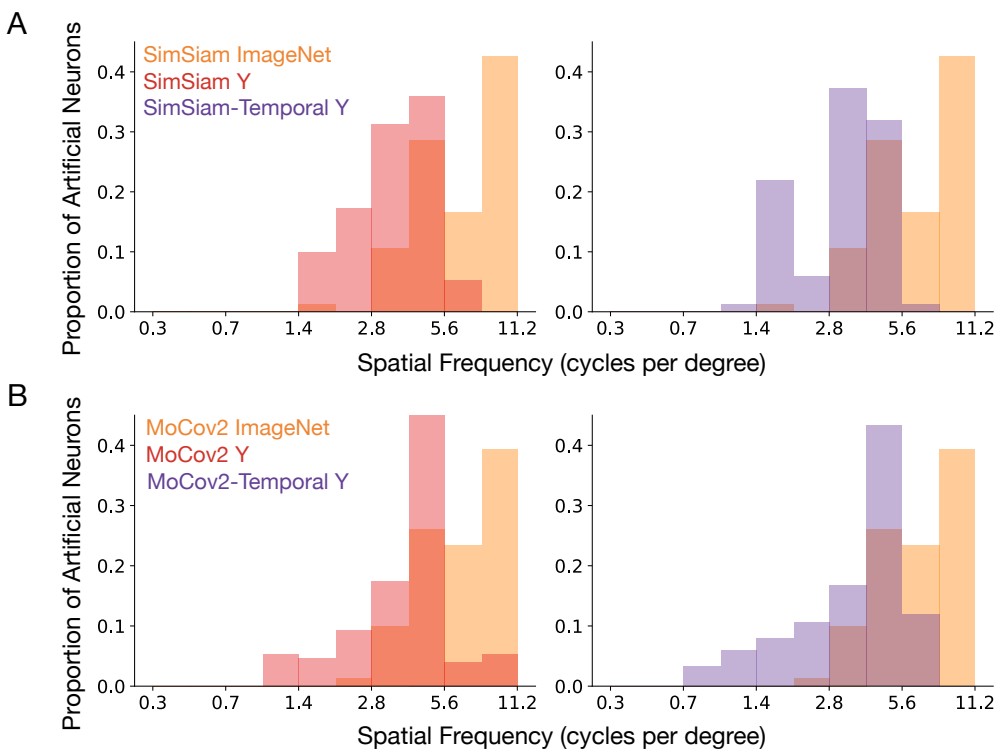

Figure 2: **Preferred spatial frequency distributions of artificial neurons in models trained on different datasets and objective functions.** The preferred spatial frequency distribution of artificial neurons in the `maxpool` layer of these models shows that models pre-trained on ImageNet in an unsupervised manner (orange) contain a substantial proportion of artificial neurons tuned to high spatial frequencies. This is in contrast to models trained on the SAYCam dataset in an unsupervised manner (red and purple), where a (relatively) larger proportion of artificial neurons are tuned to the spatial frequencies in the middle. **A**. Preferred spatial frequency distributions for models trained using the SimSiam objective. **B**. Preferred spatial frequency distributions for models trained using the MoCov2 objective.

# 4 Discussion

Current state-of-the-art computational models of the primate visual system are task-optimized deep CNNs. A closer look at these models, however, reveals fundamental ways in which human and machine perception differ. In particular, these models are notoriously brittle, where human-imperceptible image perturbations can drastically change their behavior, resulting in clear safety issues in real-world applications. Inspired by controlled-rearing studies of newborn chicks (Wood & Wood, 2016; Wood et al., 2016; Wood & Wood, 2018), we hypothesized that training models on a more temporally-continuous (i.e., naturalistic) data stream could lead to improved visual representations. In this work, we provided an initial investigation into how the training data distribution can affect a model's adversarial robustness. We found that pre-training models using infant head-cam videos resulted in higher adversarial robustness than that of models pre-trained using ImageNet and that models that were trained to treat temporally-adjacent video frames similarly (temporal contrastive learning) were more robust than models that did not use this additional temporal information. Taken together, our results suggest that adversarial robustness can be improved if models can take advantage of temporal information in the data stream and that curating datasets that are more similar to the natural image sequences infants receive could also lead to more robust models.

We first note that our results cannot distinguish between whether it is the "statistics" of the modality of video frames (e.g., higher temporal correlations) or the "statistics" of the infant visual experience *per se* that are the drivers of improved adversarial robustness. Therefore, we do not claim that infant head-cam video datasets are necessary to build models that are more adversarially robust. We also note that the adversarial attacks performed here are relatively weak, as the maximum allowable perturbation sizes presented in Table 1 are relatively small, and that adversarially training models on ImageNet still provides the most robust models. Thus, more work is needed to boost the robustness of SAYCam-trained models in regimes of larger perturbation sizes.

We suggest a few possible future directions for this work. First, we fixed the architecture to be ResNet-18 with 2D convolutions. Thus, we did not make use of the temporally-continuous nature of the SAYCam dataset architecturally (the temporal classification objective encourages models to group video frames belonging to the same time window and temporal contrastive learning treats video frames adjacent in time as positive examples). Could architectures that explicitly take advantage of the temporal continuity in video datasets learn visual representations that are more robust to perturbations? For example, 3D-ResNets have been able to achieve success in action recognition tasks (Hara et al., 2018). Can spatiotemporal filters contribute to adversarial robustness? More generally, can training on video streams improve adversarial robustness?

On the data stream front, we investigated the impact of training models on more temporally-continuous (and developmentally-realistic) inputs on adversarial robustness. This, however, can be made even more biologically plausible. In particular, infant visual acuity is known to be almost an order of magnitude lower than that of an adult (Mayer & Dobson, 1982; Norcia & Tyler, 1985). Thus, visual cortex effectively receives low-resolution inputs early in development. Could training models on blurred inputs, in concert with inputs that infants are more likely to receive, benefit adversarial robustness?

Finally, we note that in training these models, the entire dataset—video frames from six months to 30 months of age for child S—is "replayed" over and over again (once every training epoch). This is clearly not the visual experience that infants receive. Building models that are able to learn visual representations in an "online" manner would be necessary for them to be more biologically plausible and can perhaps provide insight about other parts of the brain that may be recruited to facilitate learning.

# 5 Acknowledgements

Having a large enough training dataset is one of the most important, if not *the* most important, components of building models that can mimic human capabilities and further our understanding of sensory cortex. We are therefore grateful to Sullivan et al. (2020) for their Herculean efforts in collecting and in making available the longitudinal infant head-cam video dataset. We are also grateful to A. Emin Orhan for providing initial assistance with the SAYCam dataset.

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

## A   Appendix

### A.1   Examples of annotated video frames from child S

The set of annotated video frames from child S were obtained from Orhan et al. (2020). Figure 3 shows four samples from three of 26 categories derived from those video frames. These samples were obtained from the validation set, which was used to evaluate previously-trained 26-way linear classifiers and the adversarial accuracy of the models.

### A.2   Examples of adversarial images

Figure 4 shows two adversarial examples generated using a model pre-trained on ImageNet in a supervised manner.

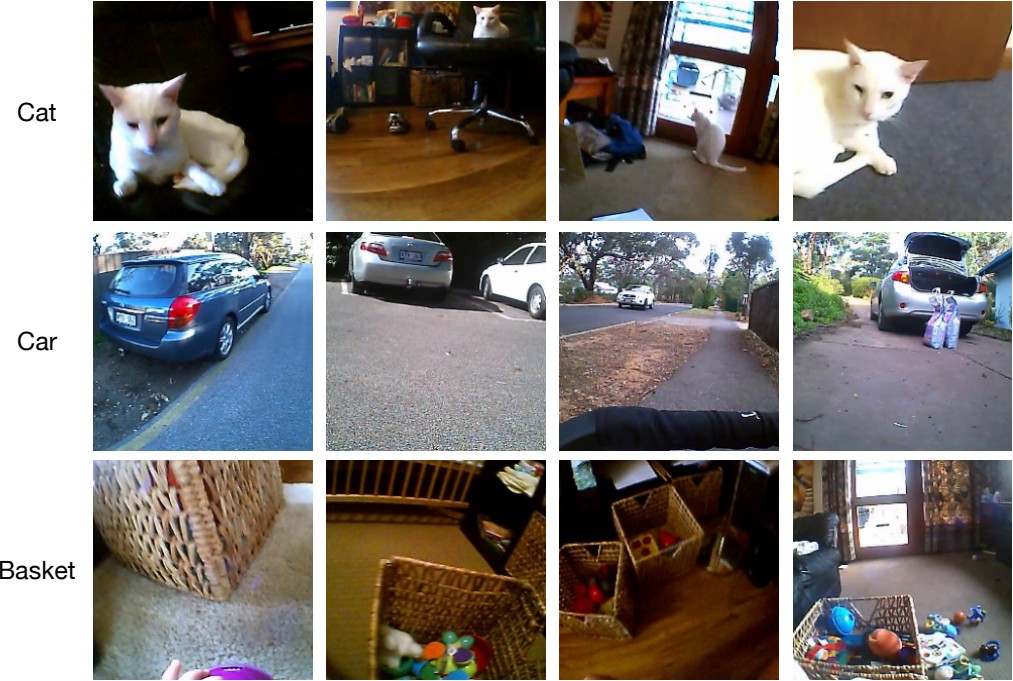

Figure 3: **Examples from the set of annotated video frames of child S.** Samples from three of the 26 categories in the set of annotated video frames of child S obtained from Orhan et al. (2020) are shown.

### A.3 Preferred spatial frequency tuning of supervised ImageNet models

Training (self-supervised) models with Gaussian blur image augmentations could have led to reduced sensitivity to high spatial frequencies (shown in Figure 2). Figure 5 shows that training models on ImageNet in a supervised manner with more image augmentations (the same as those used in self-supervised algorithms) does not decrease the proportion of artificial neurons tuned to high spatial frequencies relative to that of a supervised ImageNet model trained without Gaussian blur image augmentation (and also without other augmentations used in self-supervised algorithms).

### A.4 Additional model training details

Here we report additional details of model training and provide the image augmentations used via APIs belonging to the `torchvision.transforms` package.

**Temporal classification**  Models trained on temporal classification were trained for 20 epochs using the Adam optimizer (Kingma & Ba, 2014), a batch size of 512, weight decay of $1 \times 10^{-9}$, and an initial learning rate of $1 \times 10^{-3}$ that was decayed by a factor of 10 at epochs 10 and 15. Two GPUs were used to train each of these models. The following image augmentations were used:

```
transforms.Compose([
    transforms.RandomResizedCrop(224, scale=(0.2, 1.)),
    transforms.RandomApply(
        [transforms.ColorJitter(0.9, 0.9, 0.9, 0.5)],
        p=0.9
    ),
    transforms.RandomGrayscale(p=0.2),
    transforms.RandomApply(
        [GaussianBlur(sigma_min=0.1, sigma_max=2.0)],
        p=0.5
```

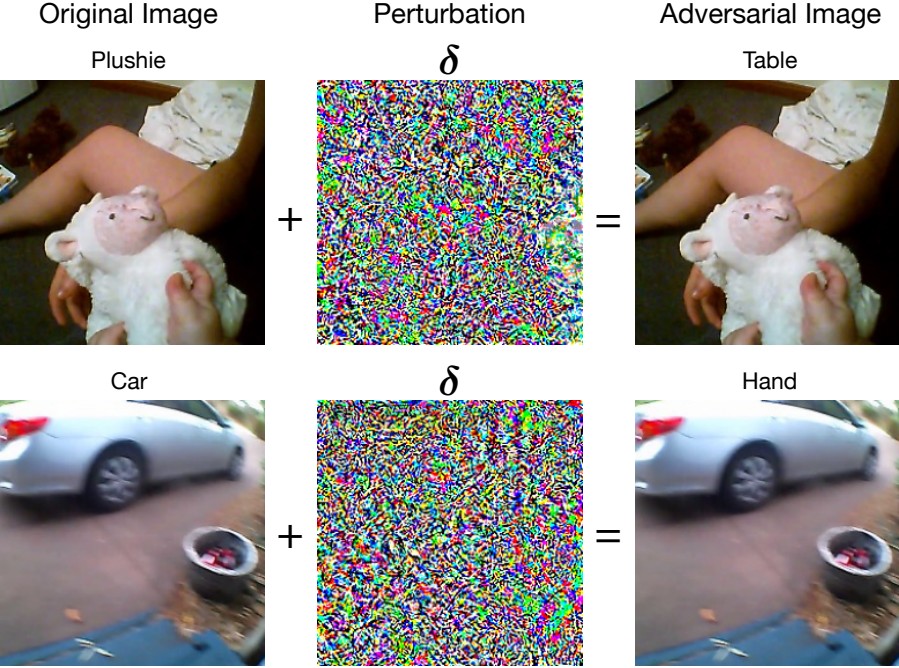

Figure 4: **Human-imperceptible image perturbations can drastically affect model predictions.** The left column shows the original images along with their correct model predictions. The right column shows their corresponding adversarial images and their incorrect model predictions. The middle column shows the perturbations (with $\|\boldsymbol{\delta}\|_\infty \leq 1/1020$) that were added to the original images to obtain the adversarial images. Note that their values have been magnified so that the perturbations are visible. The adversarial images were generated from the annotated video frames of child S, using a model pre-trained on ImageNet in a supervised manner ("Supervised ImageNet").

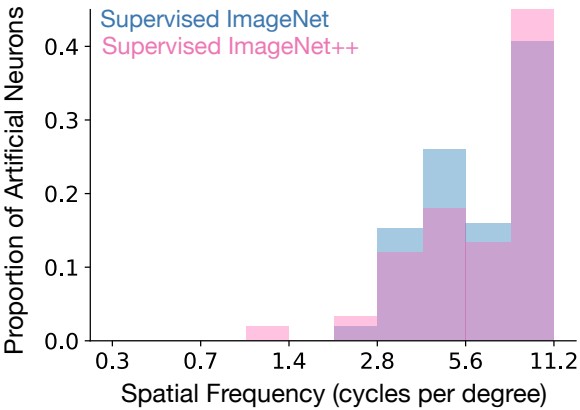

Figure 5: **Preferred spatial frequency tuning of supervised ImageNet models trained with different image augmentations.** Both models trained on ImageNet in a supervised manner exhibit a large proportion of artificial neurons in the `maxpool` layer preferring high spatial frequencies. "Supervised ImageNet++" was trained with more image augmentations than "Supervised ImageNet", including random Gaussian blur.

```
    ),
    transforms.RandomHorizontalFlip(0.5),
    transforms.ToTensor(),
    transforms.Normalize(
        mean=[0.485, 0.456, 0.406], std=[0.229, 0.224, 0.225]
    ),
])
```

**SimSiam**   Models that were trained using SimSiam and the SAYCam dataset were optimized for 20 epochs using stochastic gradient descent (Bottou, 2010) with momentum of 0.9, a batch size of 256, weight decay of $1 \times 10^{-9}$, and an initial learning rate of 0.1 that was updated using a cosine annealing schedule. Two GPUs were used to train each of these models. The model that was trained using SimSiam and ImageNet was optimized for 100 epochs using stochastic gradient descent with momentum of 0.9, a batch size of 512, weight decay of $1 \times 10^{-4}$, and an initial learning rate of 0.1 that was updated using a cosine annealing schedule. Four GPUs were used for training this model. The following image augmentations were used (identical to those of Chen & He, 2021):

```
transforms.Compose([
    transforms.RandomResizedCrop(224, scale=(0.2, 1.0)),
    transforms.RandomHorizontalFlip(),
    transforms.RandomApply(
        [
            transforms.ColorJitter(
                brightness=0.4, contrast=0.4, saturation=0.4, hue=0.1
            )
        ],
        p=0.8,
    ),
    transforms.RandomGrayscale(p=0.2),
    transforms.RandomApply(
        [GaussianBlur(sigma_min=0.1, sigma_max=2.0)],
        p=0.5
    ),
    transforms.ToTensor(),
    transforms.Normalize(
        mean=[0.485, 0.456, 0.406], std=[0.229, 0.224, 0.225]
    ),
])
```

**MoCov2**   The model that was trained using MoCov2 and ImageNet was optimized for 200 epochs using stochastic gradient descent with momentum of 0.9, a batch size of 512, weight decay of $1 \times 10^{-4}$, and an initial learning rate of 0.06 that was decayed according to a cosine annealing schedule. The model that was trained using MoCov2 and the SAYCam dataset was optimized for 12 epochs using stochastic gradient descent with momentum of 0.9, a batch size of 256, weight decay of $1 \times 10^{-9}$, and an initial learning rate of 0.03 that was updated using a cosine annealing schedule. Two GPUs were used for training these models. The image augmentations used were the same as those used by models trained using SimSiam (described above).

**Temporal contrastive learning**   Models trained using SimSiam and MoCov2 were also trained using variants of them. On top of using image augmentations as positive examples, each video frame's immediate neighbors (in time) were also treated as positive examples (Orhan et al., 2020). For the model trained using MoCov2, the number of training epochs was 12, the initial learning rate was 0.03 and was decayed according to a cosine annealing schedule, the batch size was 256, the momentum was 0.9, and the weight decay was $1 \times 10^{-9}$. For the models trained using SimSiam, the number of training epochs was 15, the initial learning rate was 0.1 and was decayed according to a cosine annealing schedule, the batch size was 256, the momentum was 0.9, and the weight decay was $1 \times 10^{-9}$. Two GPUs were used for training these models. The image augmentations used were the same as those used by models trained using SimSiam and MoCov2 (shown above).

**Supervised ImageNet++**  A model was trained on ImageNet in a supervised manner using image augmentations used by the contrastive learning algorithms, instead of the generic augmentations used during training of the ResNet-18 model from PyTorch's model zoo (i.e., random resized crop, random horizontal flip, and normalization). This model was optimized for 100 epochs using stochastic gradient descent with momentum of 0.9, a batch size of 256, weight decay of $1 \times 10^{-4}$, and an initial learning rate of 0.1 that was reduced by a factor of 10 at epochs 30 and 60.

**Linear classifier training**  In most cases, the linear readout weights were trained for 100 epochs to minimize the cross-entropy loss using the Adam optimizer, a batch size of 256, weight decay of 0.0, and a constant learning rate of $5 \times 10^{-4}$. The linear readout for the "Random" model was trained for 100 epochs with an initial learning rate of $1 \times 10^{-3}$ that was decayed by a factor of 10 at epoch 75. For most models that were pre-trained using the SimSiam and MoCov2 objectives, the linear readout weights were trained for 200 epochs with a constant learning rate of $5 \times 10^{-4}$. The linear readout for the "SimSiam S" model was trained for 200 epochs with an initial learning rate of $1 \times 10^{-3}$ that was decayed by a factor of 10 at epoch 120. The linear readouts for the "SimSiam-Temporal S" and the "SimSiam-Temporal Y" models were trained for 200 epochs with a constant learning rate of $1 \times 10^{-3}$. The linear readout for the "Supervised ImageNet++" model was trained for 100 epochs with a constant learning rate of $5 \times 10^{-4}$. A single GPU was used to train the linear classifier for each model. The following image augmentations were used:

```
transforms.Compose([
    transforms.RandomHorizontalFlip(0.5),
    transforms.ToTensor(),
    transforms.Normalize(
        mean=[0.485, 0.456, 0.406], std=[0.229, 0.224, 0.225]
    ),
])
```

## A.5  Adversarial perturbation size constraints

Here we provide additional details on the size constraints applied to the adversarial perturbations, $\boldsymbol{\delta}$. Note that the same size constraints were used by Dapello et al. (2020).

- $\ell_\infty$-norm: $\|\boldsymbol{\delta}\|_\infty \leq \varepsilon, \varepsilon \in \{0, 1/2040, 1/1020, 1/510, 1/255, 2/255, 4/255, 8/255\}$
- $\ell_2$-norm: $\|\boldsymbol{\delta}\|_2 \leq \varepsilon, \varepsilon \in \{0, 0.075, 0.15, 0.3, 0.6, 1.2, 2.4, 4.8\}$
- $\ell_1$-norm: $\|\boldsymbol{\delta}\|_1 \leq \varepsilon, \varepsilon \in \{0, 20, 40, 80, 160, 320, 640, 1280\}$

