# OpenReview forum: "Are models trained on temporally-continuous data streams more adversarially robust?"
_NeurIPS.cc/2021/Workshop/SVRHM — SVRHM 2021 Poster_

### Official Review · Reviewer_A7ec · 2021-10-22
**Interesting idea, but could use more direct comparisons**

**Rating:** 6
**Confidence:** 4

**Review:**

The authors study the impact of continuous video data on adversarial robustness. This is a very interesting idea exploring a potential biological mechanism on robustness, as most ML models are trained on highly randomized samples, a setting in which authors note animals do not learn well and perform significantly worse on classification tasks. As a first attempt at testing the impact of temporally-continuous data, this is a very interesting article. The authors demonstrate that the ResNet-18 trained on this continuous dataset achieves higher robustness than an imagenet trained model for small perturbation sizes. While the paper is clearly written, there are several comparisons and sanity checks missing that would make this a much stronger work, listed below:

- The authors claim that the effect of choosing the training set on robustness has not been studied much. However, one can claim that the whole approach of adversarial training, in which models are trained on adversarial examples is such an endeavor.
- It would have been good to compare not only the SAYCam to imagenet models, but crucially to adversarially trained models, perhaps on both datasets.
- While the imagenet model was restricted to 26 classes, it is not clear whether the choice of classes had an impact on the actual performance of the model - this should have been tested.
- More importantly, it is not clear at all whether one can compare adversarial accuracy between two different datasets. What would most likely be a better comparison would be to scramble up the frams in the SAYCam dataset to eliminate the temporal continuity and compare to a model trained on that. The fact that the SimSiam ImageNet model had a very similar curve to that of a random network with a trained classifier suggests that the ImageNet trained models did not transfer well to the child S dataset.

These above points, especially the last one cast some doubts over how honest the comparisons were, but nonetheless this is an interesting first attempt.

---

### Official Review · Reviewer_UZvj · 2021-10-29
**Interesting topic, interesting results and well written. Limited novelty and incremental contribution.**

**Rating:** 7
**Confidence:** 4

**Review:**

This paper studies the adversarial vulnerability of models pre-trained on a more naturalistic data set (SAYCam, head-cam video on infants) with a temporal classification objective, under the hypothesis that such learning scenario should yield better visual representations. The authors compared these models with others pre-trained on ImageNet classification, ImageNet with contrastive learning and contrastive learning on SAYCam. While all models are largely vulnerable to weak adversarial attacks, the results indicate that models trained on ImageNet classification are the least robust and models trained on SAYCam with temporal classification are more robust.

This paper is well written and addresses an interesting topic that I would deem relevant for the audience at SVRHM. The experiments are well designed and the results are clearly explained. I positively value that the authors compared various models to allow draw clearer conclusions, and I also appreciate the experiments to measure the spatial frequency selectivity of the trained models. As another positive aspect, I highlight that the authors identify some of the limitations of their work at the end of the paper, as well as set the grounds for possible future directions, with which I agree. For these reasons, I have a general positive impression of the paper and I recommend its acceptance for presentation at SVRHM. Nonetheless, I have some concerns and questions that I discuss below.

First, I have one comment regarding the motivation for the work: the authors mention as primary motivation the work by Wood and Wood (2016) and Wood et al. (2016), who "found that newborn chicks reared with more temporally continuous visual experience were better able to generalize to novel viewpoints of objects and also had better object recognition abilities than chicks reared with less temporally continuous visual experience". While these are very interesting results and arguably relevant for the fields of computational neuroscience and deep learning, I believe that the connection with adversarial vulnerability is not direct. Perhaps a more direct measure of the impact of training with more naturalistic data sets and learning objectives would be the invariance of the learned representations, as in Hernandez-Garcia et al. (2019) or more recently Biscione and Bowers (2021). What do you think?

Second, I think it is fair to mention in the review that the novelty of this paper seems to be limited to the analysis of the adversarial vulnerability of a set of models from an experimental setup that is nearly identical to that in Orhan et al. (2020), which the papers cites in multiple instances as a reference for the methodological decisions.

Third, while the results do shed some light on the impact of the data set on the adversarial vulnerability of the models, which is the objective set by the authors, I have the impression that the results leave more questions than answers. First, all models including those pre-trained on SAYCam on a temporal classification task are largely vulnerable to adversarial examples, performing not much better than the random baseline. This weakens, in my opinion, the significance of the conclusions. Second, we see that the main outlier in the results (Table 1) is _Supervised ImageNet_ (row 1), which performs significantly worse than any other model on adversarial examples. Already SiamSiam ImageNet is less vulnerable than the model trained with cross-entropy. Therefore, a fair questions is whether it is really the data set that matters for adversarial vulnerability or the learning objective? The answer is probably a complex one that will require future work. I would like to read more discussion about these questions, ideally in the paper and in context with the recent relevant literature.

Finally, a more technical comment is that the presentation of the results could be further improved, in my opinion, if they were presented graphically rather than on table, in order to enable easier comparisons.

### References

* Hernandez-Garcia et al. [Learning robust visual representations using data augmentation invariance](https://arxiv.org/abs/1906.04547v1). 2019.
* Biscione and Bowers. [Learning online visual invariances for novel objects via supervised and self-supervised training](https://arxiv.org/abs/1906.04547). 2021.

---

### Official Review · Reviewer_aQ8V · 2021-10-30
**Child-Like Visual experience improves robustness to adversarial perturbations**

**Rating:** 4
**Confidence:** 4

**Review:**

This work focuses on the effect of the visual experience (dataset) on the robustness of neural networks to adversarial attacks. In Particular, they used a SAYCam dataset model to replicate the visual experience of a human child.


Pros:

- Focuses on the effect of temporal continuity and data statistics on adversarial robustness.
- Compares several models, particularly ImageNet trained models and SAYCam trained models.
- Interesting results relating spatial frequency preference and depending on the training dataset.

Cons:

- It is not clear if the Supervised ImageNet trained model used the same data augmentation techniques as SimSiam and the Temporal Classification Models. This could be a problem because data augmentation has been shown to improve robustness to adversarial attacks.
- Spatial frequency preferences plots show that both SimSiam and TC models have a higher preference for lower spatial frequency compared to the Supervised ImageNet model. In the supplementary material, they showed that SimSiam and TC models were trained using Gaussian blurring, which could bias the preference towards lower frequency information.
- Does not include some of the control models of previous works (Orhan et al, 2020). For example, single frame SAYCam trained models. This as suggested by the authors could help disentangle between the effect of modality statistics and dataset statistics on their robustness results.
- Accuracy vs Maximum allowable size of perturbation curves do not include confidence intervals so it is hard to evaluate gain in robustness in certain curves.

Recommendations:

- Produce confidence intervals for the Accuracy vs Maximum allowable size of perturbation curves.
- Test adversarial robustness on single frame SAYCam trained models.
- Compared against a Supervised ImageNet trained model that had the same data augmentations as TC and SimSiam models.

---

### Decision · Program_Chairs · 2021-11-02

Accept (Poster)